# Do Spinal Needle Sizes Affect the Development of Traumatic CSF in Neonatal LP Procedures?

**DOI:** 10.3390/children10030509

**Published:** 2023-03-04

**Authors:** Aysen Orman, Hilal Aydın

**Affiliations:** 1Departments of Neonatology, School of Medicine, Mersin University, Mersin 33110, Turkey; 2Departments of Pediatric Neurology, School of Medicine, Balikesir University, Balikesir 10145, Turkey

**Keywords:** newborn, lumbar puncture, traumatization, desaturation

## Abstract

Lumbar puncture (LP) is widely employed to evaluate infectious, neurological and metabolic diseases in the newborn. Neonatal LP is a difficult procedure with 45–54% success rates. Although there are studies examining traumatic LP failure, studies on the effects of needle sizes are limited. This study was intended to investigate the effect of needle sizes on LP traumatization. Term and premature babies who underwent LP in the neonatal intensive care unit between 30 November 2017 and 30 July 2019 were included in the study by retrospective file scanning. LP was performed by a pediatric or neonatal specialist using a 22 Gauge pen (G) or 25 G pen spinal needle in all cases, with all patients being placed in the lateral decubitus position. The primary outcome was to evaluate the effect of needle sizes used in LP on traumatization. The secondary outcome was to evaluate traumatization rates and complications. A statistically significant difference was determined in the rate of traumatized LP and desaturation development between needle sizes and CSF microscopic findings (*p* = 0.031, *p* = 0.005, and *p* = 0.006, respectively). The study data show that 25 G pen-tip spinal needles cause less traumatic LP in neonates than 22 G pen-tip spinal needles.

## 1. Introduction

Lumbar puncture (LP) is a procedure performed to obtain a specimen of cerebrospinal fluid (CSF) for cytological, biochemical, or microbiological analysis [1]. Symptoms such as fever, seizure, persistent vomiting, fontanelle swelling, and irritability constitute indications for LP in the neonatal period [2]. It is frequently used in the evaluation of infectious, neurological and metabolic diseases [3,4,5]. It is also one of the most frequent invasive clinical procedures performed by pediatricians [6]. However, neonatal LP is a difficult procedure with 45–54% success rates [7,8]. An unsuccessful LP or uninterpreted CSF result can lead to a risk of exposure to complications such as delayed or missed diagnosis, incorrect antibiotic therapy, repeated LPs, and potential long-term sequelae [8,9,10]. Successful LP is, therefore, extremely important [11]. The safety and success rate of the LP procedure are the two main reasons for concern. While these aspects have been subjected to considerable investigation in adults and children, the data for neonates with lower gestational ages are more limited [12,13,14] Studies of risk factors for traumatic or failed LP are limited. The incidence of traumatic LP in the emergency department has been reported to be significantly lower than that for LPs performed in other clinical settings in case series involving adults [15]. Factors that have been linked to a heightened risk of traumatic LP in children older than one month include clinician experience, use of local anesthetics, presence of stylets, and increased patient mobility [13]. Although LP traumatization rates are 15–50%, they differ according to the health centers concerned [3,4,5,16]. However, risk factors for neonatal traumatic LP have not to date been defined categorically. Neonatal LP is probably performed less frequently than recommended [11,17]. Clinicians may be reluctant to administer LP to newborns because they fear adverse events, or find it technically difficult or overly invasive. The probability of LP being performed seems to diminish as birth weight decreases [17,18,19]. A study from the USA noted that even in the case of proven early-onset sepsis, 80% of term babies and 40% of preterm babies underwent LP [17].

Based on the relationship between needle size and spinal hematoma, it has been hypothesized that the incidence of traumatic LP in neonates can be lowered by means of a needle with a smaller external diameter [11]. Contemporary expert neurology and anesthesia practice opinion has suggested that needle size is a potential affecting factor [20,21]. Although there are studies examining LP failure in the neonatal period, research into the effects of needle sizes in LP is limited. The aim of this research was to investigate the effect of needle sizes on traumatization in LP procedures.

## 2. Materials and Methods

### 2.1. Study Design, Setting, and Population 

The study was carried out between 30 November 2017 and 30 July 2019 in Turkey’s Cengiz Gökçek Children’s Hospital Neonatal Intensive Care Unit (NICU) by means of a retrospective file review. The hospital’s NICU has 83 incubators (56 level 3, 17 level 2, and 10 level 1) and a nurse-patient ratio of 1/3. The study was approved by the institutional clinical research ethics committee (7 December 2022, 2022/803).

#### 2.1.1. Case Definition and Inclusion Criteria

Term and preterm infants who underwent LP during the study period (with indications such as meningitis, sepsis, neonatal convulsion etiology, or metabolic disease diagnosis) were eligible for inclusion in the study. Preterm babies were defined as those born at <37 gestational weeks (GW), and term babies (early term and term) as those born at 37–42 GW. Only the first LP performed was recorded. Neonates subjected to LP prior to admission to the NICU, with ultrasound-proven intraventricular bleeding (IVH) ≥ grade 2, without LP, with contraindications for LP (thrombocytopenia, presence of infection at the lumbar puncture site, hemodynamically unstable babies, or spinal anomalies), or undergoing repeated LPs were excluded. The classification developed by Volpe for IVH was used based on cranial ultrasonography images [22]. Grade 2 and 3 IVH and periventricular hemorrhagic infarction represent contraindications for LP. Term and preterm newborns with IVH at this grade were excluded from the study.

#### 2.1.2. Data Collection 

The demographic characteristics of the infants enrolled in the study (sex, week of birth, body weight, age (days) when LP was performed), and clinical characteristics (LP indication, sizes of needles used to perform LP [22 or 25 gauge], desaturation during LP [saturation < 92% and bradycardia], transfontanelle.

USG results for evaluating complications after LP, CSF laboratory results (glucose, protein, cell count and culture, traumatic LP (>500 erythrocytes per mm)] were analyzed using Statistical Package for Social Sciences version 22 (SPSS 22) retrospective file scanning. Based on the previous literature, traumatic LP was defined as a CSF RBC count > 500 erythrocytes per mm^3^ [8,23]. An increased white blood cell count (>30 cells/mm^3^, sensitivity and specificity 80%), increased protein concentration (>150 in preterms, >100 mg/dL in term babies), decreased glucose concentration (<20 mg/dL in preterms, <30 mg/dL in term infants), or a CSF value below 70–80% of the concomitant blood glucose value were regarded as abnormal CSF findings [24].

#### 2.1.3. Lumbar Puncture 

LP was performed using a standard sterile technique, with all patients being placed in the lateral decubitus position by a senior nurse. LPs were all performed either by the same neonatal specialist or by the same pediatrician. The neonatal specialist had 11 years of professional experience, and the pediatric specialist had nine years. Topical lidocaine and intravenous paracetamol were administered before LP. Blood glucose was measured immediately before the procedure for simultaneous evaluation with CSF glucose. All punctures were performed from the L4–L5 interval [25,26]. Pen-tipped 22 gauge (G) or 25 G needles were used. According to the formula of spinal needle depth [0.03 × length (cm)], the needle was advanced approximately 1–1.5 cm in term infants and 1 cm in preterm infants [27]. The stylet was removed to see whether the fluid was emerging. If no fluid was observed, the spinal needle was rotated. Samples 0.5–1 mL in volume were placed into sterile tubes for biochemical examination, microscopic evaluation, culture and other optional tests (CSF metabolic tests, PCR). Successful LP procedures were defined as those in which a suitable CSF sample was obtained and red cell count analysis was performed.

#### 2.1.4. Outcome Measure 

The primary outcome was to evaluate the effect of the needle sizes used in LP on traumatization. The secondary outcome was to evaluate traumatization rates and complications. The diagram of the cases that underwent LP is shown in Figure 1.

### 2.2. Statistical Analysis

Descriptive variables were first investigated in both independent groups. The normality of the distribution of continuous variables was assessed using the Kolmogorov–Smirnov and Shapiro–Wilk tests. Since the continuous variables were not normally distributed, the Mann–Whitney U-test was applied to compare these between the two groups. Qualitative data were expressed as the absolute reference and percentage distributions, and quantitative variables as mean and standard deviation in case of normal distribution and as median (minimum–maximum) values otherwise. The χ^2^ test was applied to compare categorical variables. The results were analyzed on Statistical Package for Social Sciences version 22 software (SPSS, Chicago, IL, USA), and *p* values < 0.05 were considered significant.

## 3. Results

Forty-six (46.9%) of the patients were preterm (<37 weeks), and 53.1% (*n* = 52) were at term between 37 and 42 gestational weeks. Fifty-two cases (53.1%) were male, and 46 (46.9%) were female. The patients’ mean age was 11.16 ± 7.89 (3–32) days, the mean gestational age was 36.08 ± 2.84 (27.20–40) weeks, and the mean body weight (birth weight) at the time of LP was 2585.05 ± 766.52 g (650–5100 g) (Table 1).

Fifty-four (55.1%) LPs were performed by pediatricians, and 44 (44.9%) by neonatal specialists. Fifty-three (54.1%) LPs were performed with a 22 G pen-tipped needle and 45 (45.9%) with a 25 G pen-tipped needle. The traumatic LP rate was 12.2%. While no cells were found at CSF microscopic examination in 66.3% of the cases, leukocytes were >30 mm^3^ in 13 (13.3%) cases. Growth in CSF culture was detected at a rate of 6.1%. Meningitis was diagnosed in 21.4% of cases. Additionally, 18.4% of the cases were desaturated during LP. LP findings by needle sizes are shown in Figure 2.

Indications for LP were sepsis (*n* = 53, 54.1%), metabolic disease (*n* = 18, 18.4%), hydrocephalus etiology (*n* = 15, 15.3%), and neonatal seizure etiology (*n* = 12, 12.2%). While the findings of transfontanelle ultrasonography (TFUSG) were normal in 70.4% (*n* = 69) of the cases, the most common abnormal findings were communicating/non-communicating hydrocephalus (*n* = 15, 15.3%), and the second most frequent finding was hypoxic changes/leukomalacia (*n* = 7%, 7.1%). The least frequently detected abnormal finding was grade 1–2 intracranial hemorrhage (*n* = 1, 1%).

Anti-convulsant treatment was applied to 24 (24.5%) cases and broad-spectrum antibiotic and/or antifungal treatment to 53 (54.1%), while 21 (21.4%) cases were followed up without treatment or metabolic support for diagnosis. The demographic, clinical and laboratory characteristics of the cases that underwent LP are presented in Table 1.

Statistically significant associations were found between needle sizes and CSF microscopic findings and rates of traumatized LP and desaturation development (*p* = 0.031, *p* = 0.005, and *p* = 0.006, respectively).

The characteristics indicating the relationship between the demographic, clinical and laboratory characteristics of the patients who underwent LP and the needle sizes are given in Table 2.

## 4. Discussion

The distinctive features of this study are that statistically significant associations were observed between the needle sizes used during LP and CSF microscopy findings and traumatization and desaturation rates.

Although LP is frequently performed on neonates, lower success rates have been reported compared to other patient populations [14]. The most frequent complication during LP is traumatization. Traumatic LP is frequently seen in the neonate population, with incidences in previous studies ranging from 35% to 46% [3,28]. Previous studies have reported traumatic LP rates between 10% and 30%, depending on the clinical setting, the age of the patient and the RBC count threshold employed [8,15,29,30,31]. Greenberg et al. [16] reported that 39.5% of 6374 LPs were traumatic, and meningitis was also detected in 50 newborns with traumatic LP. The reported incidence of traumatic LP for ≥500 erythrocytes u/L is 42.9% in newborns and 22.5% in infants [32]. The incidence of traumatic LP is inversely proportional to the patient’s age and the experience of the physician performing the procedure. A prospective study reported a higher risk of traumatization and failure in 1459 children who underwent LP without anesthesia compared to LPs performed with local anesthesia [33]. Bedetti et al. [34] reported an LP failure rate of 38.2%. Desaturation was also observed more frequently during LP in infants with a lower gestational age in that research [34]. Procter et al. [35] reported a failure rate of 32.3% in 350 patients who underwent LP, compared to 30.7% in those receiving sedation, and concluded that sedation could reduce failed LP rates. A large retrospective study of pediatric oncology patients cited black ethnicity, a younger patient age, a low platelet count, and a lower level of practitioner experience as risk factors for traumatic LP [35]. The use of local anesthetic and early removal of the spinal needle stylet have also been reported to increase success rates among infants under three months of age (12 weeks) [12,36]. In recent years, many studies in pediatric anesthesia and oncology have promoted the use of intravenous analgesia and anesthesia for a successful LP [37,38,39]. Factors affecting the success of the procedure include the age of the patient, the experience of the clinician carrying out the procedure, the needle being advanced with the stylet in place, the type of stylet employed, the use of topical anesthesia, and the position in which the patient is placed [13,28,36,40]. Researchers are agreed that factors such as a patient’s age of less than one year, not using a local anesthetic, delayed stylet removal, and increased movements on the part of the patient can result in failed LP [9,13,36]. However, several studies indicate no association between the LP position and the level of training of the practicing clinician (physician or intern), the body weight of the infant, or corrected gestational age in preterm infants [3,28,41,42] In the present study, all patients were administered topical lidocaine and intravenous parasetamol before LP. The physicians practicing LP were trained and experienced. All patients were placed in the lateral decubitus position by a senior staff nurse. The traumatized LP rate was 12.2%. The traumatic LP rate with 22 G pen-tipped needles was 11.2% but was statistically significantly lower, at 1% with 25 G needles. Based on these data, it may be concluded that traumatic LP can be reduced with a 25 G pen-tipped spinal needle when potential causes of LP traumatization defined in the literature such as the experience of the operator, the LP poisiton, and sedation are brought under control.

The complexity of neonatal units and the fact that they provide healthcare to a vulnerable population further add to clinicians’ concerns over patient safety. Adverse events should therefore be recorded during and after the intervention in order to be able to perform safe invasive procedures. Previous studies have evaluated the stability of vital signs, desaturation, and bradycardia development as basic safety indicators during LP [42]. Research has considered the safety of LP in young infants in terms of the severity of desaturation, mostly monitored using pulse oximetry [40,43,44]. However, due to the limited number of these studies, the question of whether the desaturations are associated with the LP procedure itself or with the infant’s position during it remains a controversial one [40,45]. In the present study, LP was performed with patients in the lateral decubitus position only, and 18.4% of cases were desaturated during LP. Therefore, no comparison was possible with different LP positions. However, while the desaturation rate was 15.3% during LP application with a 22 G pen-tipped needle, the equivalent rate with a 25 G pen-tip needle was 3.1%, and the difference was statistically significant. In conclusion, the 25 G pen-tipped LP needle caused less desaturation among infants undergoing LP.

In terms of previous studies concerning ideal needle sizes for LP, Carson and Serpell [46] examined needle types and sizes, arguing that time is of primary importance for accurate measurement of the CSF flow rate and CSF pressure. Those researchers defined the ideal minimum CSF flow rate as 2 mL/min and the ideal needle size as 20 G. However, they recommended not using a needle larger than 22 G and accepting a flow below the optimal CSF flow rate in order to reduce patient morbidity [46]. They showed that the atraumatic 22 G needle could deliver a CSF flow three times higher (1.6 mL/min) than that of the 22 G traumatic needle. Based on these results, individualized needle selection depending on the LP indication may be recommended, using a 22 G needle to measure CSF pressure and a smaller diameter needle for CSF samples only [46,47,48]. The larger the diameter of the needle used for LP, the greater the risk of traumatization and postdural puncture headache (PDPH) [47,48,49]. Due to the anatomical and physiological characteristics of newborns, it appears likely that the complication rate will rise in line with the size of the needle (>22 G). Reducing the needle diameter will lower the fluid flow rate due to the resulting increase in pressure. The slow CSF flow in smaller-diameter atraumatic needles may therefore result in early needle withdrawal or LP being inaccurately evaluated as unsuccessful. The present study compared the 22 G pen-tip spinal needle currently present in our unit and the 25 G pen-tip spinal needle. Smaller-diameter pen-tip spinal needles (26 G, 27 G, or 29 G) were not used since these were not available in our unit.

Previous studies of spinal needle sizes have considered postdural headache in children and have reported that a smaller diameter needle (27 G vs. 26 G) caused less dural traumatization and reduced PDPH [50]. A similar study suggested that a 25G pen-tipped needle can be successfully employed for diagnostic/therapeutic LP in pediatric oncology patients and reported a significant decrease in the incidence of back pain and a rather minimal trend toward a shorter duration of PDPH symptoms [51]. Headache appears to decrease due to the reduction in dural traumatization with smaller needle sizes in both studies. Other rare but described complications in neonates include transient neurological manifestations (dysesthesia and nerve palsies), local infection and bleeding, cerebral herniation, CSF leakage into the epidural space, and spinal epidermoid tumors (associated with needle insertion without a stylet). Most of these complications are mild, but some require prompt diagnosis and treatment [25,26,52,53]. Considering all these complications and traumatization rates, it is important to investigate the extent to which spinal needle size can exhibit an effect in newborns. Grade 2 IVH was detected at TFUSG performed after LP using a 25 G pen-tipped needle in a 12-day-old infant born at 27 + 2 GW 650 g. It was unclear whether this hemorrhage was a post-LP complication or a comorbidity associated with prematurity. No complication capable of being linked to LP was detected in any of the other patients in this study. Although studies of LP needle sizes in newborns are limited, the most recent research was the prospective observational study by Flett et al. [11], which included 250 neonatal cases. In that study, 56.8% of patients underwent LP with 22 G and 43.2% with 25 G. Those authors observed a statistically significant relationship between needle size and LP repetition, traumatic CSF, and smaller CSF samples being sent to the laboratory due to needle size [11]. Flett et al.’s [11] study is important as the first to evaluate the effect of needle sizes (25 G vs. 22 G) on the incidence of traumatic LP. Findings consistent with that study were also observed in the present research. Spinal needle sizes were similar, with 54.1% of LPs being performed with 22 G (*n* = 53) and 45.9% with 25 G (*n* = 45). Lower traumatization and desaturation rates were detected with the smaller spinal needle (25 G).

Another subject investigated was the amount and the flow time of the CSF obtained using LP. The use of 25 G needles increases the duration of the procedures due to the slower CSF flow compared to 22 G needles. Crock et al. [54] reported that the fluid collection time doubled using a thinner needle (25 G). A 25 G needle will exhibit an approximately 8.5-fold decrease in flow compared to a 22 G needle using Hagen-Poiseuille’s law for the laminar flow of a Newtonian fluid through an extended cylindrical tube with a constant cross-section [11]. To summarize, the CSF flow rate will accelerate as the needle diameter decreases. The amount of CSF sent to the laboratory in this study was sufficient for clinical and laboratory diagnosis. Although this suggests that needle sizes have no effect on the amount of CSF sample collected, we were unable to arrive at a definite conclusion about the effect of needle sizes since we did not record the amount of CSF (mL) and the flow time.

In Bedetti et al.’s [34] study of 204 newborn babies, 65.7% were term infants, and 34.3% were preterm. In Guo et al.’s [55] study, 57.9% of 171 newborns were boys, and 42.1% were girls. The mean gestational age was 22.5.0 ± 23.4 days, the mean birth weight was 1411.04 ± 431.5 g, and the mean postnatal age was eight days at the time of LP. In the present study, 46.9% of the cases who underwent LP were female, 53.1% were male, the mean gestational age was 36.08 ± 2.84 (27.20–40) weeks, the mean body weight was 2585.05 ± 766.52 (650–5100) g, and the mean age was 11.16 ± 7.89 (3–32) days. The preterm rate was 46.9%, and the term rate was 53.1%.

The incidence of bacterial meningitis is higher in the first month of life than at any other time [56]. Sadly, neonatal meningitis still results in significant morbidity and mortality, despite all the progress recently made in neonatal intensive care [57]. The first signs of meningitis may be indefinite, particularly in newborns, although a swollen anterior fontanelle, nuchal rigidity, and opisthotonus may develop within a matter of hours [57]. LP in the neonatal period is, therefore, often used to confirm whether or not concomitant meningitis is suspected in sepsis. Sepsis, bacteremia and purulent meningitis are the most frequently observed indications of LP in newborns [55]. In the present study, the most frequent indications for LP were sepsis (54.1%), suspected metabolic disease (18.4%), and hydrocephalus etiology (15.3%). Meningitis was determined in a quarter of our cases (24.7%). This figure is indicative of how important it is for physicians to be able to perform LP successfully using a safe procedure without hesitation.

The limitations of this study include its retrospective and single-center nature and the fact that due to the limited patient numbers, we were unable to separate the needle sizes according to gestational weeks (advanced preterm, preterm, and term). Our data may differ from those of centers where LP is carried out under different conditions. In addition, this study compared 22 pen G and 25 pen G spinal needles, and the efficacy of small spinal needles was not investigated. Maranhao et al. [58] compared the effect of needle groups on PDPH, a particularly undesirable complication of spinal anesthesia, in adults. The lowest likelihood of post-dural puncture headache and unsuccessful procedures in that study was achieved with 26 G atraumatic needles [58]. Had spinal needles with a diameter of 26 G or less been available in our unit, then it would have been possible to compare the effect on traumatization and desaturation in newborns. Another limitation may be that LP positions were not compared. However, LP was performed with the patient only in the lateral decubitus position for reasons of standardization and in order to be able to evaluate the effect of needle sizes on traumatization more clearly. However, Guo et al. [55] reported that LP performed with the patient in the prone position may be safer for premature and low birth weight newborns. The ideal future study might prospectively randomize term and preterm neonates to LP using ideal LP positions and matching needle sizes based on practitioner experience. Further prospective studies involving the participation of multiple centers and larger case numbers are now needed.

## 5. Conclusions

In conclusion, the 25 G pen-point spinal needle appears to entail a lower prevalence of desaturation and traumatization than the 22 G pen-point spinal needle in newborns. We believe that our results will pave the way for randomized controlled studies in preterm and term infants because of the difficulty in achieving successful LP in newborns as well as its importance in diagnosis and treatment.

## Figures and Tables

**Figure 1 children-10-00509-f001:**
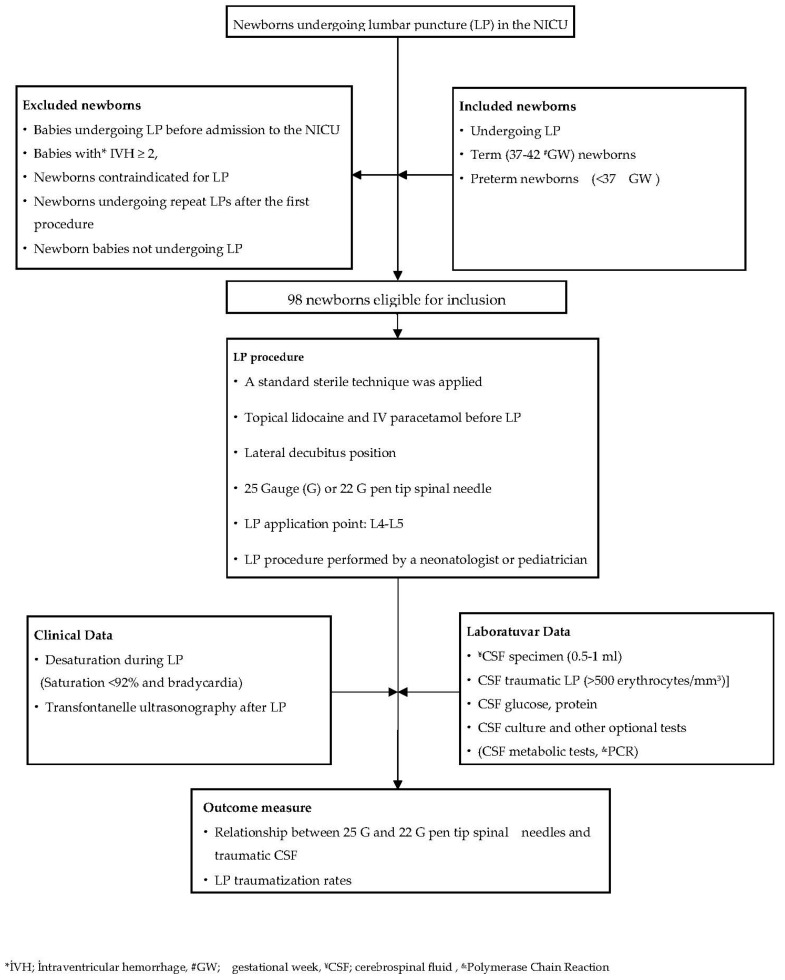
Flow chart of patient selection and the methods employed.

**Figure 2 children-10-00509-f002:**
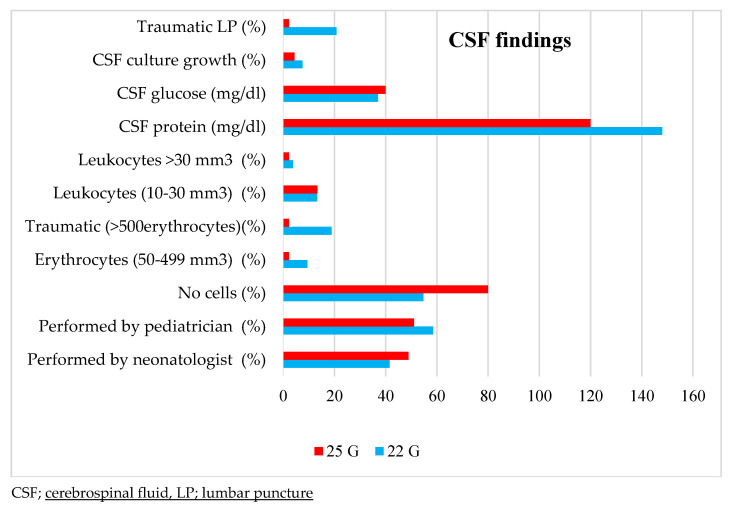
LP findings by needle sizes.

**Table 1 children-10-00509-t001:** Clinical and demographic characteristics of the patients who underwent LP *.

Demographic Characteristics	*n* (%)
**Gestational age (weeks)**	
<37 weeks	46 (46.9)
37–40 weeks	52 (53.1)
**Gender**	
Female	46 (46.9)
Male	52 (53.1)
**** The median age at LP (days)**	8.0 (3.0–32)
**** Gestational age (weeks)**	37.10 (27.20–40)
***** Body weight during LP (g)**	2585.05 ± 766.52 (650–5100)
**Clinical characteristics**	
**LP procedure**	
Pediatrician	54 (55.1)
Neonatologist	44 (44.9)
**LP needle sizes**	
22 Gauge pen-tip needle	53 (54.1)
25 Gauge pen-tip needle	45 (45.9)
**Traumatic LP**	12 (12.2)
**LP indications**	
Sepsis	53 54.1)
Metabolic disease	18 (18.4)
Hydrocephalus etiology	15 (15.3)
Neonatal convulsion etiology	12 (12.1)
**Desaturation during LP**	18 (18.4)
**Diagnosis of meningitis**	24 (24.7)
**Treatment**	
Anticonvulsant therapy	24 (24.5)
Broad-spectrum antibiotic and/or antifungal therapy	53 (54.1)
Treatment-free follow-up for metabolic support/diagnosis	21 (21.4)
**¥ TFUSG Findings**	
Normal	69 (70.4)
Communicating/non-communicating hydropcephalus	15 (15.3)
Cerebral malformation	3 (3.1)
Hypoxic changes/ leukomalacia	7 (7.1)
Periventricular calcification	3 (3.1)
Grade 2-3 intracranial bleeding	1 (1)

* LP; lumbar puncture, ** Median, *** Mean (minimum and maximum) values are given instead of “*n*” and “%”, ¥ TFUSG; transfontanelle ultrasonography.

**Table 2 children-10-00509-t002:** Demographic and clinical characteristics according to pen-tip spinal needle gauges (22 G and 25 G).

Demographics	22 Gauge (*n* = 53)	25 Gauge (*n* = 45)	*p*
**Median age at * LP (days)**	8.0 (3–32)	7.0 (3–31)	0.830
**Sex**			0.446
Female	23 (43.4)	23 (51.1)	
Male	30 (56.6)	22 (48.9)	
**LP procedure performed by**			0.464
Pediatrician	31 (58.5)	23 (51.1)	
Neonatologist	22 (41.5)	22 (48.9)	
# **CSF microscopic findings**			0.031
No cells	29 (54.7)	36 (80)	
Erythrocytes (50–499 mm^3^)	5 (9.4)	1 (2.2)	
Traumatic (>500 mm^3^ erythrocytes)	10 (18.9)	1 (2.2)	
Leukocytes (10–30 mm^3^)	7 (13.2)	6 (13.3)	
Leukocytes >30 mm^3^	2 (3.8)	1 (2.2)	
**** CSF protein (mg/dL)**	148 (15–2100)	120 (43.6–586)	0.736
***** CSF glucose (mg/dL)**	37.01 ± 17.95	40.44 ± 18.22	0.352
**CSF culture**			0.684
Growth	4 (7.5)	2 (4.4)	
No growth	49 (92.5)	43 (95.6)	
**Traumatic LP**	11 (20.8)	1 (2.2)	0.005
**LP indications**			0.974
Sepsis	29 (54.7)	24 (53.3)	
Metabolic disease	9 (17)	9 (20)	
Hydropcephalus etiology	8 (15.1)	7 (15.6)	
Neonatal convulsion etiology	7 (13.2)	8 (11.1)	
**Desaturation during LP**	15 (28.3)	3 (6.7)	0.006
**Diagnosis of meningitis**	13 (24.5)	11 (24.4)	0.950
**Treatment**			0.934
Anticonvulsant therapy	14 (26.4)	10 (22.2)	
Broad spectrum antibiotic and/or antifungal therapy	29 (57.7)	24 (53.3)	
Treatment-free follow-up for metabolic support/diagnosis	10 (18.9)	11 (24.4)	
¥ **TFUSG Findings**			0.725
Normal	38 (71.7)	31 (68.9)	
Communicating/non-communicating hydrocephalus	7 (13.2)	8 (17.8)	
Cerebral malformation	2 (3.8)	1 (2.2)	
Hypoxic changes/leukomalacia	5 (9.4)	2 (4.4)	
Periventricular calcification	1 (1.9)	2 (4.4)	
Grade 2-3 intracranial bleeding	0 (0)	1 (2.2)	

* LP; lumbar puncture, ** Median (minimum and maximum) and *** Mean values are given instead of “*n*” and “%”, # CSF; cerebrospinal fluid, ¥ TFUSG; transfontanelle ultrasonography.

## Data Availability

The data presented in this study are available on request from the corresponding author.

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
