# Peer review of "Do Spinal Needle Sizes Affect the Development of Traumatic CSF in Neonatal LP Procedures?"

_children, 2023, doi:10.3390/children10030509_

Round 1
Reviewer 1 Report
Thank you for permitting me to review this manuscript
here are my comments
the restrospective nature of this study should be clearly stated in the abstract and in the begining of the method section and briefly in the abstract.
If this is a retrospective study , mentionning the declaration of helsinki is unnecessary , this should be checked with the IRB
The definition of traumatic LP should be given in the etxt of method section.
Please explain xhy 27 G needle was not used for neonates, since this caliber is used in other practice such as anesthesiology as 29 G is also used , however I could imagine this would limit the amount of CSF expeled to the needle and increase the length of the procedure .
Table 1 realignement inTFUSG Findingsis necessary
Table 2 , realignement in microscopic cellular finding is necessary
Line 145 separate anticonvulsivant
Line 244 please elaborate why you think there is no relation with CS flow and needle size ,
Line 264-264 please elaborate (explain better ) or rephrase this sentence which is not clear
The authors should cite results published by other specialities neurologists and anesthesiologist on the same issue and discuss results ,
Author Response
Dear Reviewer
Thank you very much for taking your precious time to agree to review this article.
In line with your suggestions
The retrospective nature of this study should be clearly stated in the abstract and in the begining of the method section and briefly in the abstract.
Answer;
The retrospective nature of this study has been included in the abstract and method sections.
Term and premature babies who underwent LP in the neonatal intensive care unit between 30 November 2017 and 30 July 2019 were included in the study by retrospective file scanning.
If this is a retrospective study, mentioning the declaration of helsinki is unnecessary, this should be checked with the IRB
Answer;
The Declaration of Helsinki has been removed from this sentence. The IRB has been checked.
The study was approved by the institutional clinical research studies ethics committee (17.12.2022, 2022/803).
The definition of traumatic LP should be given in the etxt of method section.
Answer;
Traumatic LP was defined as a CSF RBC count > 500 erythrocytes per mm3 ( Wiswell TE, Baumgart S, Gannon CM, Spitzer AR (1995) No lumbar puncture in the evaluation for early neonatal sepsis: will meningitis be missed? Pediatrics. 95(6):803–806) ( García-De la Rosa, G.; De Las Heras-Flórez, S.; Rodríguez-Afonso, J.; Carretero-Pérez, M. Interpretation of white blood cell counts in the cerebrospinal fluid of neonates with traumatic lumbar puncture: a retrospective cohort study. BMC Pediatr 2022, 22, 1, 488,doi: 10.1186/s12887-022-03548-z.)
Please explain why 27 G needle was not used for neonates, since this caliber is used in other practice such as anesthesiology as 29 G is also used, however I could imagine this would limit the amount of CSF expeled to the needle and increase the length of the procedure.
Answer ;
In terms of previous studies concerning ideal needle sizes for LP, Carson and Serpell [46] examined needle types and sizes, arguing that time is of primary importance for accurate measurement of the CSF flow rate and CSF pressure. Those researchers defined the ideal minimum CSF flow rate as 2 ml/min and the ideal needle as a 20 G one. However, they recommended not using a needle larger than 22 G and accepting a flow below the optimal CSF flow rate in order to reduce patient morbidity [46]. They showed that the atraumatic 22 G needle can deliver a CSF flow three times higher (1.6 ml/min) than that of the 22 G traumatic needle. Based on these results, individualized needle selection depending on the LP indication may be recommended, using a 22 G needle to measure CSF pressure and a smaller diameter needle for CSF samples only (46. Carson, D.; Serpell, M. Choosing the best needle for diagnostic lumbar puncture. Neurology 1996, 47, 1, 33-37,doi: 10.1212/wnl.47.1.33. 47. Turnbull, D. K.; Shepherd, D. B. Post-dural puncture headache: pathogenesis, prevention and treatment. Br J Anaesth 2003, 91, 5, 718-729,doi: 10.1093/bja/aeg231. 48. Stendell, L.; Fomsgaard, J. S.; Olsen, K. S. There is room for improvement in the prevention and treatment of headache after lumbar puncture. Dan Med J 2012, 59, 7, A4483.)
[46-48]. The larger the diameter of the needle used for LP, the greater the risk of traumatization and PDPH [47, 49]. Due to the anatomical and physiological characteristics of newborns, it appears likely that the complication rate will rise in line with the size of the needle (>22 G). Reducing the needle diameter will lower the fluid flow rate due to the resulting increase in pressure. The slow CSF flow in smaller-diameter atraumatic needles may therefore result in early needle withdrawal or LP being inaccurately evaluated as unsuccessful. The present study compared the 22G pen tip spinal needle currently present in our unit and the 25 G pen tip spinal needle. Smaller-diameter pen tip spinal needles (26 G, 27 G, or 29 G) were not used since these are not available in our unit.
Table 1 realignement inTFUSG Findingsis necessary
Answer; The TFUSG findings in Table 1 have been restated.
Table 2 , realignement in microscopic cellular finding is necessary
Answer; The microscopic cellular findings in Table 2 have been realigned.
Line 145 separate anticonvulsivant
Line 244; please elaborate why you think there is no relation with CS flow and needle size,
Answer ;
Another subject was the amount and the flow time of the CSF obtained using LP. The use of 25 G needles increases the duration of the procedures due to the slower CSF flow compared to 22 G needles. Crock et al. reported that the fluid collection time doubled using a thinner needle (25 G). Crock, C., Orsini, F., Lee, K. J., & Phillips, R. J. (2014). Headache after lumbar puncture: randomised crossover trial of 22-gauge versus 25-gauge needles. Archives of disease in childhood, 99(3), 203-207.) A 25 G needle will exhibit an approximately 8.5-fold decrease in flow compared to a 22 G needle using Hagen-Poiseuille's law fo the laminar flow of a Newtonian fluid through a long cylindrical tube with a constant cross-section (Flett T., Athalye-Jape G., Nathan E.;Patole S. Spinal needle size and traumatic neonatal lumbar puncture: an observational study (neo-LP). Eur J Pediatr 2020, 179, 6, 939-945,doi: 10.1007/s00431-020-03580-0) To summarize, the CSF flow rate will accelerate as the needle diameter decreases. The amount of CSF sent to the laboratory in this study was sufficient for clinical and laboratory diagnosis. Although this suggests that needle sizes have no effect on the amount of CSF sample collected, we were uable to arrive at a definite conclusion about the effect of needle sizes since we did not record the amount of CSF (ml) and the flow time.
Line 264-264 please elaborate (explain better) or rephrase this sentence which is not clear
The authors should cite results published by other specialities neurologists and anesthesiologist on the same issue and discuss results
Answer ; The limitations of this study include its retrospective and single-center nature and the fact that due to the limited patient numbers we were unable to separate the needle sizes according to gestational weeks (advanced preterm, preterm, and term). Our data may differ from those of cnters where LP is carried out under different conditions. In addition, this study compared 22 G and 25 G spinal needles, and the efficacy of small spinal needles was not investigated. Maranhao et al. (Maranhao B., Liu M., Palanisamy A., Monks D. T.;Singh P. M. The association between post-dural puncture headache and needle type during spinal anaesthesia: a systematic review and network meta-analysis. Anaesthesia 2021, 76, 8, 1098-1110,doi: 10.1111/anae.15320) compared the effect of needle groups on PDPH, one of the most undesirable complications of spinal anesthesia, in adults. The likelihood of post-dural puncture headache and procedure failure was lowest with 26 G atraumatic needles (Maranhao B., Liu M., Palanisamy A., Monks D. T.;Singh P. M. The association between post-dural puncture headache and needle type during spinal anaesthesia: a systematic review and network meta-analysis. Anaesthesia 2021, 76, 8, 1098-1110,doi: 10.1111/anae.15320). Had spinal needles with a diameter of 26 G or less been available in our unit, then it would have been possible to compare the effect on traumatization and desaturation in newborns. Another limitation may be that LP positions were not compared. However, LP was performed with the patient only in the lateral decubitis position for reasons of standardization and in order to be able to evaluate the effect of needle sizes on traumatization more clearly. However, Guo et al. (Guo W., Ma D., Qian M., Zhao X., Zhang J., Liu J., Chi D., Mao F.;Zhang Y. Lumbar Puncture in the prone position for Low Birth Weight Neonates. BMC Pediatr 2022, 22, 1, 2,doi: 10.1186/s12887-021-03071-7) reported that LP performed with the patient in the prone position may be safer for premature and low birth weight newborns. The ideal future study might prospectively randomize term and preterm neonates to LP using ideal LP positions and matching needle sizes basedf on practitioner experience. Further prospective studies involving the participation of multiple centers and larger case numbers are now needed.

Reviewer 2 Report
1. Repetition of the sentence “Topical lidocaine and intravenous paracetamol were administered before LP” paragraph pages 96 to 97 and 98 -99
2. Spelling error of gestational age in table 1 “Gestasyonel” page 5
3. Spelling error of “neonatology”in table 1 page 5
4. The results are not in alignment with variables studied in the table 1 in the section “TFUSG findings”
5. In the methodology section is should be clear what is the definition of pre term
6. What is meant by grade 2 IHV ? – this should be clear in methodology section
7. In table 1 it should be grade 2 or 3 rather than stage 2 or 3 intracranial bleeding - and which grading system is used – it should be clearly referenced
8. There is a lot of emphasis in the discussion section on what the findings of other authors are, rather than focus on the current study and the significance of findings of the study
Author Response
Rewier 2
Dear Reviewer
Thank you very much for taking your precious time to agree to review this article.
In line with your suggestions
- Repetition of the sentence “Topical lidocaine and intravenous paracetamol were administered before LP” paragraph pages 96 to 97 and 98 -99
Answer: The sentence that was rewritten inadvertently has been removed from the main text
- Spelling error of gestational age in table 1 “Gestasyonel” page 5
Answer: In Table 1, the typo ın gestational age has been corrected to 'gestational'
- Spelling error of “neonatology”in table 1 page 5
Answer: The spelling has been corrected to 'neonatologist'.
- The results are not in alignment with variables studied in the table 1 in the section “TFUSG findings”
Answer: The TFUSG findings in Table 1 have been arranged to be compatible with the TFUSG findings in the results section.
- In the methodology section is should be clear what is the definition of preterm
Answer: The definition of preterm and term have been explained ın the methodology section.
- What is meant by grade 2 IHV ? – this should be clear in methodology section
Answer:
Intraventricular hemorrhage has been added to the methodology section.
The classification developed by Volpe for IVH was used based on cranial ultrasonography images [Inder TE PJV. Preterm Intraventricular Hemorrhage/Posthemorrhagic Hydrocephalus. 6th, editor. Philadelphia: Elsevier; 2018. 637-698 p.]. Grade 2, 3 IVH and periventricular hemorrhagic infarction are contraindications for lumbar puncture. Term and preterm newborns with IVH at this grade were excluded from the study.
- In table 1 it should be grade 2 or 3 rather than stage 2 or 3 intracranial bleeding - and which grading system is used – it should be clearly referenced
Answer: IVH has been corrected to grade 2-3, using the Volpe classification. This has been included in the methodology section.
- There is a lot of emphasis in the discussion section on what the findings of other authors are, rather than focus on the current study and the significance of findings of the study
Answer: The Discussion section has been revised to focused on the current study and the importance of the findings. Edits have been made to the main text.

Reviewer 3 Report
Dear authors, thanks for your manuscript.
The main question addressed by the research is to examine the effect of a smaller needle size on traumatization in neonatal LP procedures. This topic is relevant because neonatal LP is often a difficult procedure with a traumatization rate of 15-50% and there aren't specific risk factors identified. Throughout their research, the authors identify a specific risk factor in the size of the spinal needles and I hope that after this manuscript every hospital could supply smaller spinal needles to the neonatologist.
The research is well done with appropriate references; no further controls should be considered and their conclusions are correct.
Minor language editing is needed (see line 202 - intravenöz parasetamol, table 1 - Gestasyonel age, media age, etc.).
Hope this helps
Author Response
Dear Reviewer
Thank you very much for taking your precious time to agree to review this article. The manuscript has been revised with a native speaker.

Round 2
Reviewer 1 Report
The authors have responded to my queries , however I still believe the findings are not new
Author Response
Dear Reviewer
Thank you very much for taking your precious time to agree to review this article. The manuscript has been revised with a native speaker.
The manuscript draft has been uploaded in review mode with corrections.
Yours sincerely

Reviewer 3 Report
Dear authors, I appreciate your effort in language editing but there are also some errors that need to be corrected as transfontanel/transfontanelle, postdural headache (do you mean postural puncture headache?), dyesthesias... Please upload your manuscript in review mode with corrections.
Author Response

(The authors gave the same response as above.)

Round 3
Reviewer 3 Report
Dear authors, as requested before please correct some errors like media age in table 1 (median age) and transfontanelle ultrasound (brain or cranial ultrasound). Hope you understand. Thanks